# Review of Electrochemical DNA Biosensors for Detecting Food Borne Pathogens

**DOI:** 10.3390/s19224916

**Published:** 2019-11-12

**Authors:** Qiaoyun Wu, Yunzhe Zhang, Qian Yang, Ning Yuan, Wei Zhang

**Affiliations:** 1College of Food Science and Technology, Hebei Agricultural University, Baoding 071001, China; maxiaoyan@hebau.edu.cn (Q.W.); zhangyunzhe@hebau.edu.cn (Y.Z.); 2College of Science and Technology, Hebei Agricultural University, Cangzhou 061100, China; yangqian@hebau.edu.cn (Q.Y.); yuanning@hebau.edu.cn (N.Y.)

**Keywords:** food borne pathogens detection, electrochemical DNA biosensors, bioreceptors, nanomaterials, DNA amplification

## Abstract

The vital importance of rapid and accurate detection of food borne pathogens has driven the development of biosensor to prevent food borne illness outbreaks. Electrochemical DNA biosensors offer such merits as rapid response, high sensitivity, low cost, and ease of use. This review covers the following three aspects: food borne pathogens and conventional detection methods, the design and fabrication of electrochemical DNA biosensors and several techniques for improving sensitivity of biosensors. We highlight the main bioreceptors and immobilizing methods on sensing interface, electrochemical techniques, electrochemical indicators, nanotechnology, and nucleic acid-based amplification. Finally, in view of the existing shortcomings of electrochemical DNA biosensors in the field of food borne pathogen detection, we also predict and prospect future research focuses from the following five aspects: specific bioreceptors (improving specificity), nanomaterials (enhancing sensitivity), microfluidic chip technology (realizing automate operation), paper-based biosensors (reducing detection cost), and smartphones or other mobile devices (simplifying signal reading devices).

## 1. Introduction

Due to the widespread outbreak of food borne epidemics in both developed and developing countries, people are paying increasing attention to public health issue. Studies have shown that the main cause of food safety problems is food borne pathogens including *E. coli* O157:H7, *Staphylococcus aureus*, *Salmonella*, *Listeria monocytogenes*, etc. [1,2,3,4]. Food products and their raw materials are dominant transmitting agent of more than 250 known diseases [5,6]. Table 1 summarizes the relevant pathogens spread through food matrix, their main sources, virulence factors, as well as the associated epidemics. Owing to the abundance of various nutrients, raw materials are generally preferred hosts for microorganisms to grow. Although heating process can kill most of the potential bacteria in food, the emergence of ready-to-eat food in recent years increases the probability of exposure to pathogenic contamination [1]. Food products are highly susceptible to microbial contamination during processing, packaging, distribution, storage, and other stages [2]. Thus, in a modern industrial environment, the control of food processing and real-time monitoring of food borne pathogens are of paramount importance to ensure consumers’ safety.

Rapid and accurate monitoring or detection of food borne pathogens is an international priority because accurate diagnosis is one of the most effective ways to control and prevent food borne epidemics in humans and can reduce mortality rates drastically [47]. Table 2 lists various conventional methods for food borne pathogens detection, which based on microbiological methods, immunological methods, or nucleic acid-based amplification assays (e.g., polymerase chain reaction [48], loop-mediated isothermal amplification [49], rolling circle amplification [50]). Although these methods are sensitive and allowed the detection of one single or multiple bacterium, there are still several key challenges that hinder their further development from laboratory detection stage to application in market [51,52]. Summarized as the following points: time-consuming; poor sensitivity and specificity; high detection cost; non-amenable for on-site and real-time diagnosis. Among these methods, biosensor strategies are highly prevalent over other assays and fulfill the expanding demands of modern industry for detection. In addition to fast response, robustness, cost-effectiveness, high sensitivity, and selectivity, they also have the ability to detect real sample on-site with minimal sample preparation [53].

A typical biosensor combines a bioreceptor with a transducer. The bioreceptor, which specifically interacts with the target analyte, and the transducer, which converts this interaction into an electronic signal. Three basic parts of a biosensor are recognition material, transducer or detector system, and signal processor [54]. According to bioreceptors, biosensors can be classified into antibody biosensors, DNA biosensors, enzyme biosensors, whole-cell biosensors, and phage biosensor; according to transducers, biosensors can be classified into electrochemical biosensors, piezoelectric biosensors, calorimetric biosensors, and optical biosensors (Figure 1). Among various biosensors, electrochemical DNA biosensor has become a ideal alternative of conventional methods for food borne pathogens detection, due to its features such as low detection limit, wide linear dynamic range, and high reproducibility [55]. Figure 2 shows the number of annual articles published on China National Knowledge Infrastructure (CNKI) for the detection of food borne pathogens by electrochemical DNA biosensors. The literature survey demonstrates more than 31,557 articles have been published since 1986. In the past 30 years, the total number of publications has increased dramatically. The data reveals the development of electrochemical DNA biosensors in the field of food borne pathogen detection has received extensive attention. There are a number of review articles that summarize the use of electrochemical DNA biosensors to detect a specific food borne pathology, or detecting food borne pathogens by using a large class of biosensors. However, there are few review articles summarizing previous studies on electrochemical DNA biosensors for detecting food borne pathogens.

Therefore, the present review overviews extensive and up-to-date findings of electrochemical DNA biosensors for detecting food borne pathogens, summarizes the basic principle of an electrochemical DNA biosensor, DNA immobilization methods, electrochemical techniques, and detection methods. Besides, because sensitivity is a critical performance for electrochemical DNA biosensors, we also summarized several strategies for improving the sensitivity of electrochemical DNA biosensors, such as nucleic acid-based amplification technologies, which are rarely summarized in reviews. Finally, we predicted the future prospective in the field of pathogen detection.

## 2. Electrochemical DNA Biosensors

### 2.1. Basic Principle of Electrochemical DNA Biosensors

In the detection of food borne pathogens, single-stranded nucleic acids or aptamers are the preferred bioreceptor to be used in the design of electrochemical DNA biosensor. The most common transducers are gold electrodes (GE) [79,80,81], glassy carbon electrodes (GCE) [82], pencil graphite electrodes (PGE) [76] and screen printed electrodes (SPE) [83,84], and carbon ionic liquid electrode (CILE) [85]. The reaction between bioreceptor and target is performed on the electrode surface. The basic principle of electrochemical DNA biosensor is that the biological reaction between bioreceptor and target can produce or consume ions or electrons, which changes the electric current, potential, or other electrical properties of the solution. The biological signal can be converted into a detectable electrical signal proportional to target concentration by transducer and displayed on a computer [86]. The basic scheme of an electrochemical DNA biosensor is presented in Figure 3.

### 2.2. Bioreceptor of Electrochemical DNA Biosensor

#### 2.2.1. Type of Bioreceptor

For electrochemical DNA biosensors, the bioreceptor is DNA. DNA consists of two types: naturally occurring recognition element DNA and aptamer artificially synthesized in vitro with a known sequence of bases. Biosensors regarding naturally occurring DNA as bioreceptors are called ‘genosensors’. Typically, targets for such sensors are DNA of pathogens. The DNA probes immobilized on electrode surface can recognize and hybridize with targets DNA by complementary base pairing. Targets can also be recognized and captured by aptamers involving high-molecular-weight compounds, whole cells, and small molecules. Biosensors considering aptamer as bioreceptors are called “aptasensors” [86]. We attribute the high specificity of electrochemical DNA biosensors to strong affinity of ssDNA or aptamer and its target.

##### Advantages of Aptamer 

Aptamers are single-stranded DNA or RNA with 15–80 mononucleotides by artificial synthesis in vitro or peptide screened by SELEX (systematic evolution of ligands by exponential enrichment) [87,88]. The binding pattern between the aptamer and its target is similar to that of antigen–antibody, but aptamers have outstanding merits over their corresponding antibodies mainly involving in the following three points:

(1) Target diversity: It possesses the capability to bind with a wide range of targets (e.g., proteins, drugs, cell, amino acids, organic, and inorganic ions) by naturally fold into three-dimensional structures with high affinity and specificity [86,89,90,91,92]. 

(2) As amplifiable molecules: Due to the property of nucleic acids, aptamers can be amplified by polymerase chain reaction (PCR). Currently, several studies have applied this aptamer feature to improve the sensitivity of aptasensors based on real-time PCR [93,94], LAMP [95], RCA [96,97,98], et al.

(3) Low cost: Aptamers are stable even in drastic environmental conditions and not require special transport or storage conditions; aptamers can be prepared on a large scale by simple chemical synthesis with inexpensive nucleotides [99]. In addition, biosensors regarding aptamers as recognition elements have long lifetime, reducing the cost of detection.

##### Detection Mechanisms of Aptamers with their Targets

Aptamers are often considered as bioreceptors in the strategy of electrochemical DNA biosensors. When recognizing and capturing targets, aptamers fold into a three-dimensional formation. Because the elongated primary molecular structure of aptamers is unfavorable in energy and not stable, some unpaired nucleobases interact with each other to generate secondary structural motifs. The interactions of these motifs lead to more complex tertiary structures. Typical tertiary motifs are coaxial stacking and G-quadruplexes. Three different detection mechanisms are commonly applied in electrochemical DNA biosensors design for food borne bacteria detection:

(1) Direct binding mechanism: The aptamer immobilized on the electrode surface binds to target directly, causing a conformation change of aptamer, and then the current signal changes.

(2) Target-induced dissociation mechanism (TID) [88,100]: In the absence of target, the aptamer hybridizes with its complementary sequence. While the addition of target induces dissociation and replacement of the complementary sequence from the aptamer, making its complementary sequence free again and changing the electrochemical signal.

(3) Dual aptamer detection mechanism: Usually be employed in the design of sandwich-type aptasensors. In this case, the first aptamer is immobilized on sensing interface as capture probe to bind with the target and the second aptamer acts as a signal probe. 

#### 2.2.2. Immobilization Methods of Bioreceptor

The properties of electrochemical DNA sensors—including sensitivity, specificity, and lifetime—are largely related to the immobilization of bioreceptors on the surface of electrodes. Therefore, the most basic requirements for the immobilization method is that it neither destroy the biological activity of the bioreceptor nor affect the interaction between the bioreceptor and the target. Table 3 describes the principles, advantages, and disadvantages of three basic immobilizing methods of DNA.

Among various DNA immobilization methods, adsorption is the simplest, and it does not require any chemical reagents and DNA modifications. The phosphate backbone of DNA is negatively charged, and DNA can be immobilized by modifying positively charged substance on the electrode surface [101]. Such substances usually include chitosan, cationic polymeric films, etc. In addition, studies have shown that applying a positive potential to the electrode can make the DNA immobilization more stable. Vijayalakshmi Velusamy et al. [102] reported an electrochemical DNA biosensor to detect *Bacillus cereus* DNA. The gold electrode surface was modified with polypyrrole (PPy) to immobilize DNA, and then a fixed potential 0.8 V for 600 s was applied to enhance the immobilization efficiency and stability. 

Due to high efficiency of DNA immobilization and hybridization, covalent binding is one of the most common methods for DNA immobilization on electrode surfaces. Mahmoud Amouzadeh Tabrizi et al. [103] developed an electrochemical DNA biosensor based on nanoporous glassy carbon electrodes to detect *Salmonella* DNA sequences. The amino modified probe DNA was firstly covalently linked with carboxylic group on the nanoporous GCE. Then target DNA hybridized with the probe DNA and the hybridization result was obtained by DPV and EIS techniques. The LOD of the biosensor was 2.1 pM and 0.15 pM, respectively. Besides, self-assembly is another common covalent bond method for DNA immobilization on the electrode surface. Zahra Izadi et al. [76] developed an electrochemical DNA biosensor to detect *Bacillus cereus* in milk and infant formula. Before introducing DNA to the biosensor, they thiolated the 5’ end of ssDNA to immobilize ssDNA on PGE surface modified with gold nanoparticles by Au-S bonds. The biosensor sensitivity of *B. cereus* was found to be 100 CFU/mL. E. Sheikhzadeh et al. [104] established a label-free impedimetric biosensor to detect *S. Typhimurium* in apple juice. Modifying the amino group at the 5’ end of aptamer before immobilizing on electrode surface. LOD of developed biosensor was 3 CFU/mL, which achieved a satisfying detection result.

One of the most valuable strategies for the effective immobilization of DNA on electrodes is affinity binding. This method is dependent on the specific affinity between avidin and biotin. However, compared with other methods, LOD of the electrochemical DNA biosensor established by this method is low. For example, Kavita Arora et al. [105] constructed an electrochemical DNA biosensor based on polypyrrole-polyvinyl sulfonate (PPy-PVS) coated onto Pt disc electrode. DNA probe was immobilized on electrode by biotin–avidin binding (indirect immobilization) or carbodiimide coupling (direct immobilization). Compared to the indirect immobilization, the detection limit of direct immobilization of the probe increased by 2200 times and the sensitivity increased by about 6 times. In order to solve this problem, several studies have combined covalent binding and affinity binding to improve DNA immobilization efficiency. For instance, Malhotra et al. [106] developed an electrochemical DNA biosensor to detect *E. coli*. Avidin was modified with –COOH and then attached to the polyaniline (PANI)-modified Pt disk by the covalent binding between –COOH and –NH/NH_2_ of PANI. Thereby, the biotin-labeled capture probe was immobilized on electrode surface by the affinity binding. Finally, a satisfactory detection limit for *E. coli* genomic DNA (0.01 ng/uL) was obtained.

### 2.3. Electrochemical Techniques 

Due to the lower tendency for noise, voltammetry has developed into one of the most versatile electrochemical analysis techniques. The current is measured in a constant potential imposed to work electrode. Peak current intensity is proportional to the target concentration. More importantly, it is adapted to all types of bioreceptors [51]. 

Among the voltammetry techniques, cyclic voltammetry (CV), square wave voltammetry (SWV), and differential pulse voltammetry (DPV) are the most frequently employed. CV is often used to characterize chemical reactions and electrochemical coupling process on electrode surface. SWV, as a type of frequency dependent electrochemical analysis technique, is versatile and highly sensitive, which is widely used in quantitative analysis and kinetic studies of materials. Besides, the peak shape of SWV is simple and convenient for data analysis. DPV possesses the merits of lower background current and higher sensitivity, almost all electrochemical DNA biosensors regard DPV as an analysis technique for determining target concentration.

Another electrochemical technique generally applied to electrochemical DNA biosensors is electrochemical impedance spectroscopy (EIS). In this technique, impedance values are affected by the electric field changes caused by the interaction between bioreceptor and target [51]. EIS is a frequency domain measurement technique with a wide range of measurable frequencies, which allows more kinetic information and electrode interface structure information to be obtained than conventional electrochemical techniques. Table 4 listed several electrochemical techniques commonly used in electrochemical DNA biosensors for detecting food borne pathogenic bacteria.

### 2.4. Detection Methods

DNA hybridization detection methods can be classified into label-free and label-based methods [101]. For label-free electrochemical DNA biosensors, DNA’s own structure and composition are of paramount advantageous for electrochemical detection. Adenine and guanine of DNA can be easily oxidized within a certain potential range, thymine and cytosine require higher potential for oxidation, so electron transfer can be performed directly on several electrode surfaces [125,126]. Even the sensing strategy based on direct redox reaction of nucleic bases has a high sensitivity, the large background current interference limits its application [101]. The oxidation reaction of ribose destroys the phosphate backbone of DNA, it is rarely used for DNA-modified electrodes.

For label-based electrochemical DNA biosensors, numerous studies introduced redox active molecules as indicators to promote electron transfer between the electro-active base and the electrode surface. DNA was detected indirectly by measuring the electrical signals generated by the modified substances. Sensitivity of this method is much higher than that of strategy based on direct redox reaction of nucleic bases [127]. Three selection requirements for redox active indicators are summarized below. (a) Indicators cannot affect the activity of bioreceptors and cannot react with the electrode material itself. (b) Possessing the ability to bind with ssDNA or dsDNA selectively. (c) Cannot adsorb on the sensing surface. According to the above requirements, we summarized several redox active molecules applied to electrochemical DNA biosensors (Table 5).

Both Zahra Izadi et al. [76] and Wei Sun et al. [85] used methylene blue (MB) as the electroactive indicator to detect DNA hybridization efficiency, because MB can intercalated into the dsDNA structure and reaction signal was improved [54]. TB can bind to the negatively charged phosphate group of DNA. R. Nazari-Vanani et al. [80] employed this feature to examine the hybridization reaction between *Enterococcu faecalis* DNA and capture DNA. The study showed that TB binds with dsDNA in a higher extent compared to ssDNA. Thus, ssDNA and dsDNA structures on the surface of electrodes can be distinguished through the peak current. Besides, a electrochemical nanobiosensor was constructed by Mostafa Azimzadeh et al. [128] using oracet blue (OB) as an electroactive label for the first time. OB, an organic dye, has a hydrophobic rigid plane which can insert into the dsDNA base pair, causing the reduction signals to change. 

Hoechst 33258 is mainly combined with minor and major groove only existing in dsDNA. For daunomycin, its molecular carbocyclic moiety can be inserted into the base pair of the DNA helix, and its amino sugar moiety generate electrostatic interaction with the phosphate. M. Ligaj et al. [114] utilized the features of Hoechst 33258 and daunomycin to develop two electrochemical DNA biosensors for *Aeromonas hydrophila* detection. After hybridization between DNA probes immobilized on the surface of biosensors and the target DNA, the peak currents of biosensor I used Hoechst 33258 as the indictor increased by 75–135% and the other increased by 34–92%. Additionally, because Ru(phen)_3_^2+^ can intercalate into the groove of dsDNA, it is another redox indicator commonly used to detect the DNA hybridization event. For example, Huayu Huang et al. [129] selected Ru(phen)_3_^2+^ as a redox indicator to amplify electrochemical signal. Linlin Yang et al. [97] also used it to enhance the electrochemiluminescence intensity.

There are also several studies employing enzyme label-based electrochemical detection to construct direct or sandwich DNA hybridization biosensors to detect food borne pathogenic bacteria. In this method, the electrochemical signal can be amplified since the enzymes can catalyze its electroinactive substrate into electroactive products [101]. The particular redox-active enzymes typically include horseradish peroxide and alkaline phosphate. Generally, the redox-active enzymes combine with DNA by the affinity binding of biotin-avidin/streptavidin. For example, a sandwich DNA hybridization genosensor was developed for *Salmonella* detection [130]. Interestingly, this study used AuNPs-horseradish peroxidase-streptavidin (AuNPs-HRP-SA) as the signal tag to amplify the detection signal. SA biofunctionalized HRP bind to the biotinylated DNA by affinity. In the presence of H_2_O_2_ and hydroquinone (HQ), the electrochemical signal was generated. Because hydroquinone (HQ) acted as a redox mediator, HRP can catalytic H_2_O_2_ to reduce. In another study, Chuang Ge et al. [100] constructed an aptasensor to ultra-sensitively determine *Salmonella typhimurium*. Streptavidin-alkaline phosphatase (ST-AP) was used as the label enzyme to combine with the biotinylated detection probe by the affinity. The electrochemical signal was improved by catalytic activity ST-AP towards enzyme substrate α-NP. Even enzyme label-based electrochemical detection is used widely, high cost of enzyme production and the instability of enzymes limit its further development [101].

Besides those, nanoparticles commonly act as the reporter labels to characterize the DNA hybridization. Most of the previous studies applied this approach to sandwich genosensors. In the sensors, target DNA will co-hybridize with both of capture probe immobilized on the electrode surface and signal probe labeled with nanoparticles. For instance, a sandwich electrochemical genosensor using cadmium sulfide nanoparticles (CdSNPs) as a label was established by Mandour H. Abdalhai et al. [79] to detect *Escherichia coli* O157:H7. The signalizing probes were modified by amine (NH_2_) and then attached to CdSNPs by covalent binding. Finally, the sandwich structure immobilized on the electrode surface was dissolved in HNO_3_ solution, then the released CdSNPs ions were ready for electrochemical measurement. Although the method is highly sensitive, the detection method, including chemical synthesis of nanoparticles, is extremely cumbersome. 

## 3. Strategies for Improving the Sensitivity of Electrochemical DNA Biosensors

Sensitivity is a critical performance for electrochemical DNA biosensors. When detecting targets with low concentration, the reaction signal is difficult to distinguish from the background signal due to the small amount and the limitation of method itself, resulting in inaccurate detection results. Thus, in the past few decades, researchers have made enormous efforts into integrating biosensors with other disciplines such as nanotechnology and molecular biology technology to improve the sensitivity. Herein, we summarized various nanomaterials and DNA amplification technologies employed widely to improve the sensitivity of electrochemical DNA biosensors.

### 3.1. Nanomaterials

Recently, in view of their intrinsic properties—such as high surface area to volume ratio, great electronic conductivity, and excellent physico-chemical properties, carbon nanomaterials (containing carbon nanotubes, carbon nanofibers, graphene, nanoparticles et al.)—as a class of promising candidate for biosensing material, have successfully been applied to sensing strategies to improve the sensitivity of electrochemical DNA biosensors [47]. Figure 4 shows several nanomaterials of different dimensions (0D, 1D, 2D, 3D) commonly used in electrochemical DNA biosensors for food borne pathogen detection. 

#### 3.1.1. Conventional Nanomaterials

Reduced graphene oxide (RGO), gold nanoparticles (AuNPs) and carbon nanotubes (CNTs) are the three most common nanomaterials modified on electrodes. 

Graphene has large specific surface area, outstanding biocompatibility and highest mechanical strength in known materials. However, many studies introduced RGO to electrode surface, because graphene has the irreversible agglomeration, which limited its application [82,132,133,134]. The abundant oxygenated groups (hydroxyl, epoxy, carbonyl, and carboxyl groups) of graphene oxide (GO) reduce the electron transporting ability of the electrodes. So many studies employed several appropriate reduction methods to remove these oxygenated groups of GO [135]. 

Due to its small particle size, special stability and catalytic properties, AuNP is considered as the most favorite nanomaterial in the field of biosensors [135,136]. AuNPs provide substrates for DNA immobilization on the electrode surface, because DNA can be immobilized on the surface of the AuNPs-modified electrode by Au-S bonds. Besides, AuNPs can maintain the biological activity of DNA and enhance the capacity of DNA immobilization and hybridization [85]. Many studies have combined RGO and AuNPs to improve electrode performance synergistically [85,107,135,137].

Since its discovery in 1991 [138], CNTs have received extensive attention in the field of electrochemical sensors due to its mechanical flexibility, rapid electron transfer, excellent electrochemical stability, and unique thermal conductivity. Additionally, CNTs are easily modified by various functional groups and bind with DNA, and so can be widely used in the construction of electrochemical DNA biosensors [139].

#### 3.1.2. Composite Nanomaterial

Safiye Jafari et al. [140] constructed an electrochemical DNA biosensor based on Ceria nanoparticles decorated reduced graphene oxide (CeO_2_NPs-RGO) to detect *Aeromonas hydrophila* DNA. Polyaniline and CeO_2_NPs-RGO were used to modify the glassy carbon electrode. PANI provided a large surface area for DNA immobilization, and it can interact with RGO by Π–Π stacking. RGO was also used for the DNA immobilization through Π–Π stacking between its conjugated interface and DNA bases. Besides, CeO_2_ possessed high catalytic activity and biocompatibility, and can adsorb DNA through electrostatic attraction. Thus, in this study, ssDNA was immobilized on the GCE surface without any functionalization or mediators. [Ru(bpy)_3_]^2+/3+^ redox signal was used as electrochemical marker and square wave voltammetry was used as detection technique. Finally, a wide linear range of 1 × 10^−15^–1 × 10^−8^ mol L^−1^ and a low LOD of 1 × 10^−16^ mol L^−1^ were obtained. The biosensing strategy can be applied to DNA detection in other area such as clinical diagnosis, food safety, and environmental monitoring.

Yange Sun and colleagues took advantage of chitosan-multiwalled carbon nanotubes (CS-MWCNTs) and gold nanoparticles (AuNPs) to modify an Au electrode (AuE) [141]. Chitosan own the merits of nontoxic nature, excellent film forming ability, and cost effectiveness. However, the electrical conductivity of CS is very poor. In this study, MWCNTs were utilized to dope into CS film to improve the electrical conductive properties of CS, because MWCNTs have the advantages of high electrical conductivity, chemical stability, and a high surface-to-volume ratio. AuNPs can provide a platform for DNA immobilization and can enhance the electrical conductivity with MWCNTs synergistically. The thiol-modified ssDNA was immobilized on the AuNPs/CS-MWCNTs/AuE by Au-S bonds and then hybridized with the target DNA. Methylene blue (MB) was used as an electrochemical indicator. In the end, this platform reached a detection limit of *Staphylococcus aureus* DNA as low as 3.3 × 10^−16^ M and the linear detection range was 1.0 × 10^−15^–1.0 × 10^−8^ M.

In addition to the above proposed composite nanomaterials used in electrochemical DNA sensors, Table 6 lists several other nanocomposites applied in the sensing design to synergistically enhance the analysis performance of biosensors.

#### 3.1.3. Emerging Nanomaterials 

In recent years, several emerging nanomaterials were introduced to the design of electrochemical DNA biosensors for food borne pathogens detection. The emergence of these novel nanomaterials provides more choices for improving the sensitivity of electrochemical DNA biosensors and has greatly promoted the development of nanotechnology in the field of electrochemistry. 

A electrochemical DNA biosensor was reported to detect *Enterococcus faecalis*, which based on a new gold nanostructure of ice crystals-like as the sensing substrate [80]. The nanostructure provided a suitable substrate for DNA immobilization and improved the rate of DNA hybridization. ssDNA was self-assembled on the Au/nano electrode surface by thiol-gold covalent bonds. Toluidine blue (TB) was used as the DNA hybridization indicator. The DNA biosensor can detect target DNA with a outstanding LOD of 4.7 × 10^−20^ mol L^−1^.

In recent years, some studies have combined other novel nanomaterials with GO to improve the sensitivity simultaneously. Yan Li et al. developed an electrochemical DNA biosensor to detect *E. coli* O157:H7 *eaeA* gene based on a novel sensing tag of GOx-Thi-Au@SiO_2_ nanocomposites [144]. In this study, the combination of Au@SiO_2_ and GO can not only enhance electronic transfer, but it also offers a microenvironment to maintain the DNA conformation and free them in orientation. DNA was immobilized on the electrode surface by the Au-S bonds between Au@SiO_2_ and SH-DNA. Due to these merits, the biosensor demonstrated a wide linear response for *E. coli* O157:H7 *eaeA* gene in the range from 0.02 to 50.0 nM and the lowest detection limit was 0.01 nM. The developed biosensor can also apply to the detection of other pathogens with excellent performance.

### 3.2. Nucleic Acid-Based Amplification Technologies

Nucleic acid-based amplification technology is another promising alternative for electrochemical DNA biosensors to improve the sensitivity in the detection of food borne pathogens. The electrochemical DNA biosensors that integrated the powerful amplification capability of DNA amplification technology, the high sensitivity of electrochemical assays and various signal amplification strategies, are widely used in the ultrasensitive detection of trace targets. Here are several amplification methods commonly used in the construction of electrochemical DNA biosensors (Figure 5).

#### 3.2.1. Target Cycle Amplification Technique

##### Exonuclease III-Assisted Target Cycle Amplification

Exonuclease III is a kind of enzyme which catalyze the stepwise removal of mononucleotides from 3′ terminus of dsDNA in the case of substrate with a blunt or recessed 3′-terminus. Compared with the endonuclease, exonuclease III is sequence-independent and can amplify a signal without a specific recognition site of target DNA, which has been widely applied in nucleic acid detection [117]. 

A sensitive electrochemical sensing methodology for quantitative detection of *Enterobacteriaceae* bacteria was proposed by Caihui Luo et al. [117]. As shown in Figure 6, in this strategy, thiol-modified capture probes were firstly immobilized on the gold electrode surface by Au-S bonds. MCH was used to block non-specific adsorption sites on the electrode surface. In the presence of a target, it hybridized with capture probes to generate dsDNA. The dsDNA possessed unique characteristic 3′-blunt end at the capture DNA and 3′-overhang end at target DNA. Hereafter, Exo III recognized and digested the phosphodiester bonds of the 3′ end of capture probes. Then target was released to perform the next hybridization and cleavage cycle. After finishing the entire amplification, the capture probes left on the electrode surface hybridized with biotinylated detection probes. ST-AP was indirectly linked to detection probe through the affinity between biotin and streptavidin, and then enzymatic electrochemical signal was produced. By DPV technique, the proposed biosensor lead to a superior LOD of 8.7 × 10^−15^ mol/L toward *Enterobacteriaceae*. The strategy was successfully applied to the detection of *Enterobacteriaceae* in milk with a ultra-low detection limit of 40 CFU/mL.

According to the similar principle, Qianqian Pei et al. constructed a universal DNA biosensing platform to detect *S. Typhimurium* ultra-sensitively [123]. The uniqueness in their works that a duplex DNA probe was ingeniously designed by hybridizing a *Salmonella typhimurium* aptamer with a primer. In the presence of *Salmonella typhimurium*, the three-dimensional conformation of aptamer changed to bind with targets. Then primer was released and Exo III-aided amplification reaction was initiated. Another difference is that MB was used to enhance the electrochemical signal in this platform. An excellent LOD of 2.8 × 10 CFU/mL was obtained by DPV, which is lower than those of the previously proposed assays.

##### Circular Strand-Replacement Polymerization

Circular strand-replacement polymerization (CSRP) performs an isothermal amplification process under the action of DNA polymerase by utilizing a stem–loop DNA as template and target DNA as trigger. Firstly, target DNA hybridizes with loop structure, opening the stem–loop DNA, and resulting in exposure of the primer binding region. In the presence of primer, dNTPs, and DNA polymerase, the primer extends forward along DNA probe and displaces target DNA. The displaced target DNA triggered another round of primer extension and strand displacement. CSRP has been widely used to signal amplification strategies since it does not require any specific recognition sites.

Based on the CSRP principle, Ting Wang et al. developed an ultrasensitive electrochemical DNA biosensor to detect *mecA* gene of methicillin-resistant *Staphylococcus aureus* [146]. From Figure 7, we can observe hairpin probes modified with MB were firstly immobilized on gold electrode surface by Au-S bonds. MB is an electrochemically active molecule, a large current will be generated when it is restricted close to the electrode surface. When targets were present, the stem–loop probes were opened to expose the primer binding area and its loop area hybridized with targets. In the action of dNTPs and polymerase, the primers extended along probe DNA to displace the targets. The released targets induced the next amplification. In the end, a plenty of duplex DNA complexes labeled with MB were produced. The MB molecules moved away from the electrode surface, resulting in a significant decrease in current. There was a linear relationship between the varying current and the concentration of targets. According to this strategy, a low LOD of 6.3 × 10^−14^ mol/L and a wider detection range of 7.5 × 10^−14^–2.0 × 10^−10^ mol/L were obtained by DPV. The sensitivity of this method is higher than other methods for detecting *Staphylococcus aureus*.

Fenglei Gao et al. [147] proposed an electrochemical biosensor to detection DNA hybridization by coupling CSRP with AuNPs catalyzed silver deposition on the biosensor surface. The biosensor possessed an extremely high sensitivity with a LOD of sub-femtomolar level, which attributed to the efficient amplification performance of CSRP. This approach provided a universal platform for ultrasensitive detection of DNA in biomedical and bioanalytical applications. 

##### Catalyzed Hairpin Assembly

Catalyzed hairpin assembly of DNA (CHA) requires two hairpin DNA probes H1 and H2. Target DNA can cleave the hairpin structure of H1, and the opened H1 probes further unclosed the hairpin structure of H2 probes. Since H1 and H2 have more bases than target DNA, the target DNA is replaced to trigger next displacement. CHA can be carried out without any enzyme, which reduces the detection costs greatly.

Yong Qian et al. [148] constructed a signal-on electrochemical DNA to detect DNA based on target catalyzed hairpin assembly strategy. The principle has been shown in Figure 8. Firstly, the thiolated modified beacon 1 (MB 1) was immobilized on the gold electrode surface by Au-S bonds. In the presence of targets, the hairpin probes were opened and targets hybridized to the MB 1. Then the ferrocene-labeled molecular beacon 2 (Fc-MB 2) bind with the unhybridized area of MB 1 and extended forward to displace the target DNA. The released targets induced the next amplification. Finally, a number of duplex DNA complexes labeled with Fc were generated. The current was increased dramatically because Fc was confined close to the GE surface for efficient electron transfer. Target DNA concentration can be measured by the changes in current intensity. The proposed biosensor has an excellent sensitivity with a LOD of 0.74 fM obtained by DPV. The signal-on biosensor would be applied widely because it is enzyme-free and simple to perform.

Changli Zhong et al. [149] developed an electrochemical biosensor based on hairpin assembly amplification to detect specific DNA with high sensitivity and specificity. MCH and BSA were used to block the non-specific adsorption sites jointly. The presence of DNA target initiated the hairpin assembly amplification. Eventually, the H1-H2 complex were generated. The 5′ end of biotin-modified H1 probe hybridized with capture probe immobilized on the GE surface to form capture probe-H1-H2 complex on the electrode surface. The streptavidin-alkaline phosphatase (ST-ALP) bind to biotin and catalyzed the substrate a-naphthol (a-NP) to generate electrochemical signal. Under the optimized conditions, the lower detection limit was found to be 20 pM and the linear range was 25 pM–25 nM.

#### 3.2.2. Hybridization Chain Reaction

Hybridization chain reaction (HCR) is an enzyme-free isothermal amplification technique relied on the self-assembly of two DNA hairpins [150,151], which was proposed firstly by Robert M. Dirks et al. in 2004 [152]. In HCR process, the mixture of two DNA hairpins triggers a cascade of hybridization events as the introduction of targets, which generates nicked double helices analogous to alternating copolymers [153]. In recent years, the ingenious combination of HCR and other nucleic acid-based amplification strategies for two-step signal amplification has become a research hotspot in the field of electrochemical biosensors. 

Shufeng Liu et al. proposed an isothermal, enzyme-free, and ultrasensitive biosensing strategy to detect DNA by integrated HCR and DNA catalyzed hairpin assembly (CHA) for dual signal amplification [154]. From Figure 9A, we can see a thiolated hairpin DNA probe (immobilized probe) attached to the Au electrode by Au-S bonds firstly. Accompanying the addition of target DNA, it opened the stem–loop structure and hybridized with the hairpin DNA probe. The unhybridized bases of hairpin DNA probe further opened the stem–loop area of capture probe and hybridized with it to displace the target DNA. The released target DNA carried out the successive hybridization and assembly process. After CHA, the nicked double-helix of immobilized probe and capture probe was generated. In the presence of the signal probe and auxiliary probe, HCR was triggered on the electrode surface. Since the signal probe was labeled with MB, as the reaction proceeded, more MB were immobilized on the electrode surface, which amplified the electrochemical signal largely. By the DPV measuring, the developed biosensor owned a predominant specificity and sensitivity with a LOD of 1.0 × 10^−16^ mol/L, which can be applied to other gene-related detections.

An electrochemical DNA biosensor coupling HCR to circular strand-replacement polymerization (CSRP) was constructed by Cui Wang et al. [155]. As depicted in Figure 9B, stem–loop capture DNA immobilized on the electrode surface was firstly subjected to CSRP under the induction of target DNA to generate numerous DNA duplex. The capture DNA continued to act as a template to perform HCR to product nicked double-helix. Since the two hairpin DNA involved in the HCR reaction were labeled with biotin, they can combine with streptavidin-alkaline phosphatase (ST-ALP). Then the electrochemical signal was generated by catalyzing the substrate a-naphthol (a-NP). The biosensor based on the dual amplification resulted in a high sensitivity of 8.0 × 10^−15^ mol/L and a wide dynamic range of 10 fM-1 nM.

#### 3.2.3. DNA Isothermal Amplification Technology

##### Rolling Circle Amplification

Rolling circle amplification (RCA) is an enzymatic, isothermal DNA amplification process which utilizes a circular DNA template, a single DNA primer, and Phi29 bacteriophage DNA polymerase to generate a long single-stranded DNA with multiple tandem-repeat sequences [156]. RCA is triggered by the single primer combining on the circular DNA template. The Phi29 bacteriophage DNA polymerase then elongates around the template and eventually accomplishes the circle. The amplification is successive, because the newly product strand continues to displace the previously generated strand thinks to the strand displacement activity of the Phi29 bacteriophage DNA polymerase [157]. Low temperature (30–60 ℃) requirement makes this technology attractive for electrochemical DNA biosensors and has been successfully applied to food-borne pathogens detection.

A novel electrochemical DNA biosensor was constructed for *Salmonella* detection by Dan Zhu et al., which utilized RCA and DNA-AuNPs probe for dual-signal amplification [81]. As can be seen from the Figure 10A, probe 1 was firstly immobilized on the electrode surface by Au-S bond. One end of target DNA hybridized with the probe 1. With the addition of circularization mixture containing probe 2, the other end hybridized with the probe 2 to form a typical sandwich structure. The presence of dNTPs and phi29 DNA polymerase initiated the RCA to generate micrometer-long ssDNA. Then the detection probe (biotin-DNA-AuNPs) recognized and hybridized with the ssDNA product. The streptavidin-alkaline phosphatase (ST-AP) bind to biotin and catalyzed the substrate a-naphthol (a-NP) to generate enzymatic electrochemical signal. The proposed biosensor has been applied for *Salmonella* detection in real milk sample with a prominent LOD of 6.76 × 10^−18^ mol/L.

Chuang Ge et al. [100] developed an electrochemical DNA biosensor based on RCA and target-induced aptamer displacement on gold nanoparticles to detect *Salmonella typhimurium* ultra-sensitively. From Figure 10B, we can see that AuNPs was deposited on the gold electrode to improve the electrode performance. Thiolated capture probe attached to the AuNPs by Au-S bonds. MCH was used to block non-specific adsorption sites. The sequence of capture probe was complementary hybridized with aptamer. In the presence of *S. Typhimurium*, the specific hybridization between bacteria and aptamer resulted in the release of aptamer from electrode surface. Thereby, RCA primer bind to the capture probe, leading to anchoring numerous circular templates on the electrode surface. In the presence of dNTPs and phi29 DNA polymerase, RCA was initiated to generate micrometer-long ssDNA. Biotinylated detection probe was then hybridized with the multiple tandem-repeat sequences of ssDNA. Streptavidin-alkaline phosphatase (ST-AP), as the label enzyme, combined with biotin to generate enzymatic electrochemical signal by catalyzing enzyme substrate α-NP. Finally, the sensitivity of electrochemical DNA biosensor was improved dramatically, and an outstanding LOD of 8 CFU/mL was obtained by DPV.

##### Loop-Mediated Isothermal Amplification

Loop-mediated isothermal amplification (LAMP) is a mature method of nucleic acid amplification, which was developed by Notomi et al. in 2000 [49]. LAMP performs an auto-cycling strand displacement DNA synthesis under the catalysis of DNA polymerase with high strand displacement activity. At least four primers (two inner and two outer primers) are required for amplification, resulting in a high amplification efficiency of 10^9^ copies of target DNA within an hour [122]. LAMP is especially suitable for DNA amplification of pathogens thanks to the ability to amplify a few hundred base-pair long template strands of nucleic acids and the tolerance to various inhibitors present in real food samples [158]. Several electrochemical DNA biosensors based on the high amplification efficiency of LAMP have been reported for determining food-borne pathogens.

An electrochemical DNA biosensor was established by Wei Sun et al. for *Yersinia enterocolitica* gene sequence detection [122]. Carbon ionic liquid electrode (CILE), as the working electrode, was modified with V_2_O_5_ nanobelts, multi-walled carbon nanotubes (MWCNTs), and chitosan (CTS) to form nanocomposite film. Then sequence-specific ssDNA probes were immobilized on the CILE surface. The LAMP amplicons, as the target DNA, hybridized with ssDNA probe and the hybridization reaction was monitored by the electrochemical indictor methylene blue (MB). Under the optimal conditions, by the DPV measuring, the proposed DNA biosensor showed remarkable stability and sensitivity with a LOD of 1.76 × 10^−12^ mol/L and a linear range of 1.0 × 10^−11^–1.0 × 10^−6^ mol/L. Minhaz Uddin Ahmed et al. [159] proposed a biosensing strategy to real-time monitor LAMP amplicons of *Escherichia coli* and *Staphylococcus aureus* genes by using [Ru(NH_3_)_6_]^3+^ as electrochemical indictor and square wave voltammetry (SWV) as electrochemical technique. The detection can be completed within 30 min, the LOD of *S. aureus* and *E. coli* is 30 copies/uL and 20 copies/uL, respectively. 

##### Strand Displacement Amplification

Strand displacement amplification (SDA) was proposed firstly by Walker G. T. in 1992, which based on the a strand-displacing DNA polymerase [160]. Four primers are required that possess bifunction of target recognition and endonucleases target regions. It employs a restriction endonuclease to nick target DNA and an exonuclease-deficient DNA polymerase to displace the downstream DNA strand at the nick sites. Then the displaced strands act as templates to perform an antisense DNA reaction [157]. SDA owns a high amplification efficiency of 10^8^ copies of target DNA within two hours. Thus, a large number of studies have applied this strategy to the development of electrochemical DNA biosensors to improve detection sensitivity.

Yuhua Hu et al. [119] designed a electrochemical DNA biosensor to investigate 16S rDNA of *Bacillus subtilis*, which based on target-induced strand-displacement mechanism and nicking endonuclease signal amplification. (Figure 11A). Firstly, capture probes (CP) were immobilized on the gold electrode by Au-S bonds. Ferrocenecarboxylic acid (FC)-labeled signal probe (SP) was hybridized partly with CP. In the presence of target DNA, target hybridization displaced the nine hybridized bases at the 5′terminus of SP and released the FC-modified end of SP, allowing FC to approach to the electrode surface and generate current. Then the nicking endonuclease (N.BstNBI) nick the nicking position, leading to the cleavage of CP and the release of TD. The free TD performed the second cycle of cleavage. After many cycles, remarkable FC current can be detected by the DPV in a low concentration target solution of 8.0 × 10^−17^ mol/L.

An label-free and ultrasensitive electrochemical DNA biosensor was proposed by Zhiqiang Chen et al., which integrating autocatalytic strand displacement amplification (ASDA) and hybridization chain reaction (HCR) [161]. Interestingly, ASDA strategy relied on the joint activity of nicking endonuclease (Nt.BbvCI) and Bst DNA polymerase was proposed for the first time. As can be seen from the Figure 11B, the stem–loop probe (IP) was immobilized on the electrode surface by Au-S bonds. In the presence of target, the loop part of IP was opened and hybridized with target. AS probe was designed ingeniously whose 3′ end bind to the stem part of IP. In the presence of Bst DNA polymerase and dNTPs, AS probe extend forward to displace the target. The replaced target initiated the next amplification. Thereafter, multiply dsDNA was product on the electrode surface. Then the nicking endonuclease (Nt.BbvCI) acted on the nicking position (3′ end of AS probe), leading to the release of target analogy. Under the action of Bst DNA polymerase, AS probe extend forward to form the duplex DNA structure with 5′ free end of AS probe. H1 and H2 probe were used to initiate HCR reaction to form a linear DNA concatamer with cytosine(C)-rich loop region, which can promote the in-situ synthesis of silver nanoclusters (AgNCs) as electrochemical tags for further amplified detection. In the end, with the ASDA and HCR strategies, DNA detection can be achieved with a desired selectivity and an excellent LOD of 0.16 fM. 

## 4. Summary and Conclusions

Biosensors have become ideal alternative of traditional methods and molecular detection methods for food borne pathogens detection. Especially electrochemical DNA biosensors, due to their merits of low detection limit, wide linear dynamic range, and high reproducibility. In this paper, we overviewed recent advances in the development of electrochemical DNA biosensors that applied to food borne pathogens detection. We mainly discussed from the following three aspects: food borne pathogens, electrochemical DNA biosensors and two strategies for improving sensitivity of electrochemical DNA biosensors. In the part of electrochemical DNA biosensor, we highlighted the detection principle of biosensor, main bioreceptors and immobilizing methods on sensing interface, electrochemical techniques, and detection methods. In the long term, more sensitive, specific, cost-efficient, and portable biosensors will be developed to monitor and determine food borne pathogens, thereby further controlling and preventing food borne epidemics.

## 5. Future Perspectives

Although biosensors have created a break-through and became mature increasingly in the field of food borne pathogens detection, there is still a great potential to develop the ‘ideal’ biosensors. Thus, more efficient and more economical biosensors (offering sensitivity, specificity, and portability) will be the research focuses in the future [86]. According to personal understanding, new advances of biosensors have summarized in Figure 12. 

Accuracy is the most basic requirement for the development of biosensors for food borne pathogens detection. Non-interaction of the bioreceptor with its target should be taken into consideration, because it will affect directly the accuracy of the detection result [162]. Therefore, selecting more specific bioreceptors is one of the future research priorities. For example, aptamer is widely recognized due to its advantages as detailed above. However, it is worth mentioning that the commercial development of aptamers is still in its infancy [86]. Thus, developing new bioreceptors and improving the existing bioreceptors to adapt the commercial needs are future trends. 

As the world becomes more concerned about the impact of food borne pathogens on human health, methods with high sensitivity are increasingly in demand to meet the needs of detection. A successful biosensor system should possess the capability to determine the low concentration (less than 10^2^ cfu mL^−1^) of bacterial from the complex food matrix sensitively [47]. Throughout the entire production process of biosensors, electrode modification is of paramount importance to develop biosensors with low detection limits. Thus, in the future, the emergence of novel nanomaterials such as metal–organic frameworks (MOFs) [163,164,165], nanofiber [166] will open new horizons for improving sensitivity of biosensors by increasing the surface area of electrode and accelerating electron transfer. 

Sample preparation, enrichment, and selection are vital steps for food borne pathogens detection [51]. Small sample usage, rapid analysis, and automated detection are ideal standards for the development of biosensors. Microfluidic chip technology was emerging in 1980 and became a research hotpot in miniaturized total analysis systems, which promised a novel pathway to detect DNA. The microfluidic chip has the characteristics of controllable liquid flow, minimal consumption of samples and reagents, and an improvement of analysis speed by factors of hundreds. It can perform simultaneous analysis of hundreds of samples in a few minutes or even less. It can realize the pretreatment and analysis of samples online. Therefore, in the future, the integration of microfluidic chip technology and biosensors can become a powerful tool to shorten detection time, achieving automated and high-throughput analysis of samples.

Biosensors offer an unexpected alternative to the traditional and conventional detection methods. Over recent years, enormous efforts have been made to develop biosensing technology in the field of food borne pathogen detection. However, the general cost of detection remains high, hindering its wide-scale application, especially in developing countries. So how to reduce detection costs has always been the most concerning issue for detection’s industrialization and commercialization. In order to solve this problem, paper-based biosensors may become an alternative for conventional electrodes due to its merits of low cost, simple product process, and portability. Over the long run, it is expected that future studies should focus on further improving the repeatability and sensitivity of paper-based biosensors by combining them with other signal amplification techniques.

Additionally, even biosensors have been applied widely in the detection field, expensive and large data display platforms for signal output are still required. This deficiency not only increases the cost of detection, but also is not conducive to the development of on-site detection. At present, with the ubiquity of smart-phones, several studies have utilized various mobile devices to output detection signals by developing applications. The smart-phone possesses stupendous development potential owing to its high-quality camera, feasibility, and portability. Additionally, smartphones, as reading tools, do not need any specific dedicated reading device [86]. Mohamed Maarouf Ali Zeinhom et al. designed a novel approach to read the detection result of *E. coli* O157:H7 in yoghurt and egg by a smartphone based on a field-portable fluorescent imager, which exhibited a low noise to background imaging system [167]. Similarly, Lee et al. proposed a bioassay based on smartphone-integrated dual-wavelength fluorescence to detect biomolecules, which has high accuracy and can obtain clear results [168]. However, few studies have applied smartphones or other mobile devices to electrochemical DNA biosensors, so in the future it may become a research direction for developing miniaturized and portable biosensors.

## Figures and Tables

**Figure 1 sensors-19-04916-f001:**
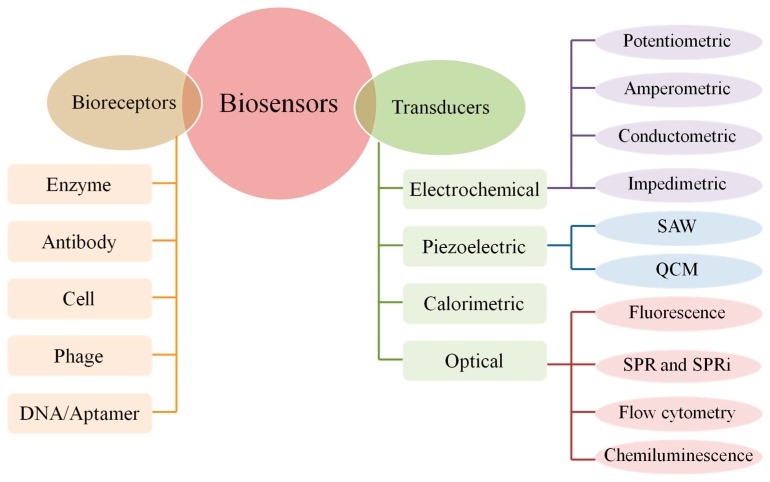
Components and classification of biosensor.

**Figure 2 sensors-19-04916-f002:**
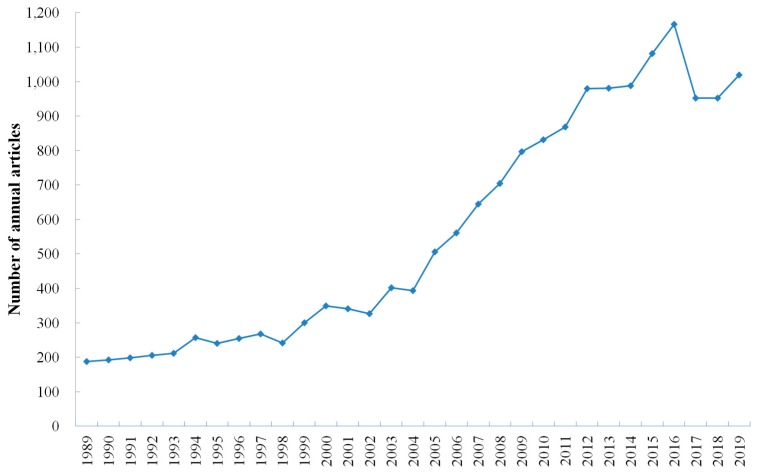
Number of publications for food borne pathogens detection with electrochemical DNA biosensors.

**Figure 3 sensors-19-04916-f003:**
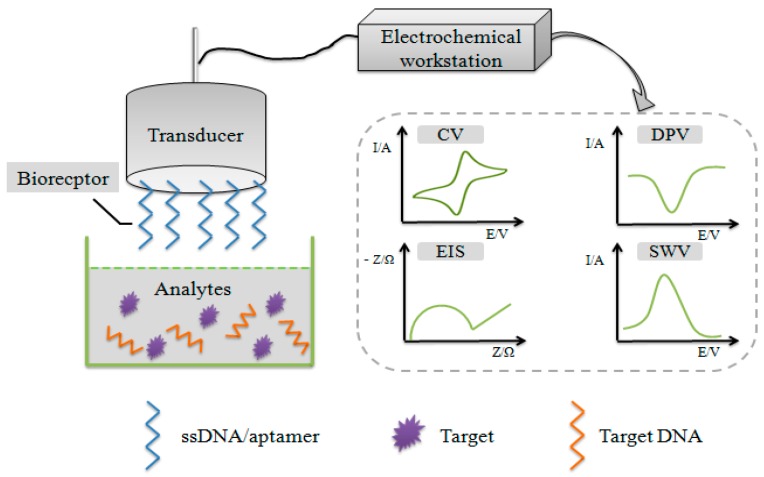
Schematic diagram of electrochemical DNA biosensors.

**Figure 4 sensors-19-04916-f004:**
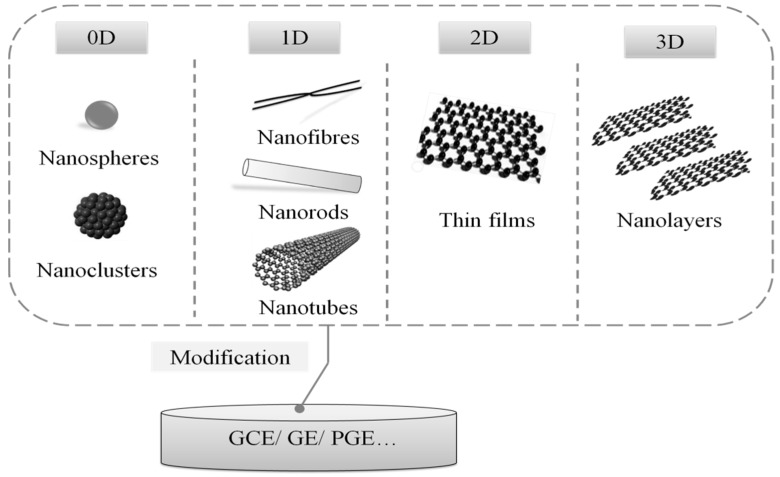
Nanomaterials commonly used in the modification of electrochemical DNA biosensors.

**Figure 5 sensors-19-04916-f005:**
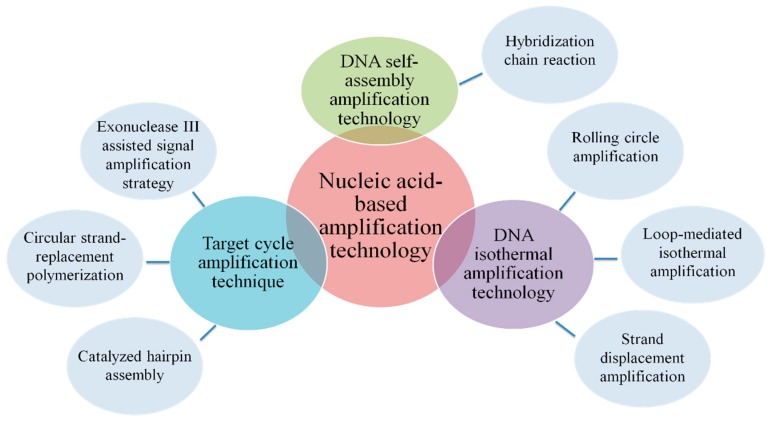
Classification of nucleic acid-based amplification technologies.

**Figure 6 sensors-19-04916-f006:**
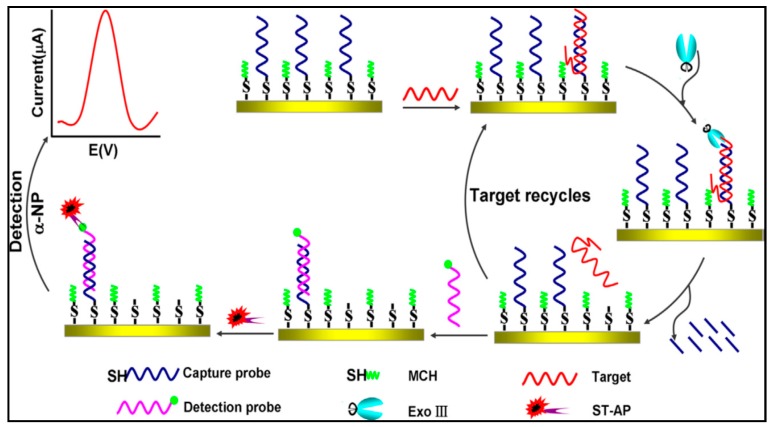
Electrochemical sensing methodology based on exonuclease III-assisted target recycling amplification technique for quantitative detection of *Enterobacteriaceae* bacteria [117]. Copyright 2013. Reproduced with permission from Elsevier B.V.

**Figure 7 sensors-19-04916-f007:**
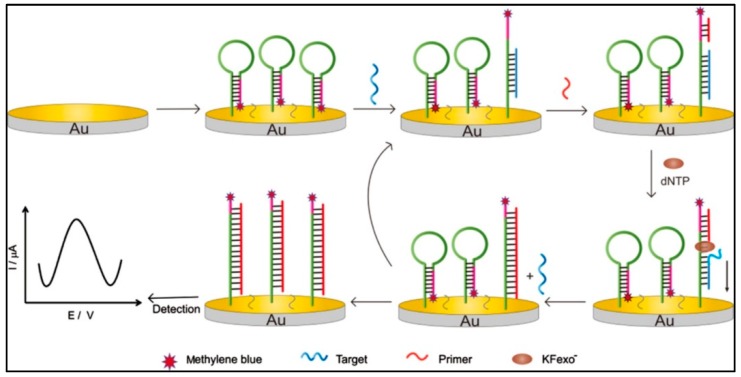
Electrochemical DNA biosensor based on CSRP to detect *mecA* gene of methicillin-resistant *Staphylococcus aureus* [146]. Copyright 2015. Reproduced with permission from Elsevier B.V.

**Figure 8 sensors-19-04916-f008:**
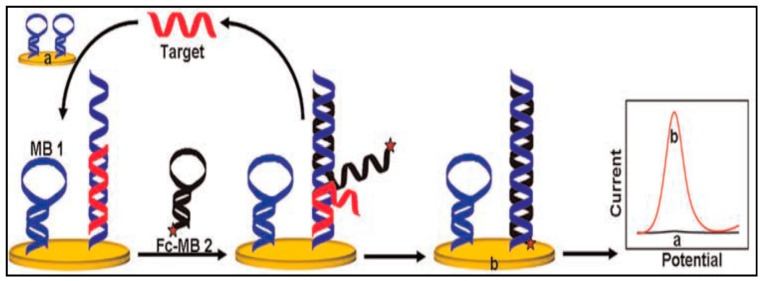
A signal-on electrochemical DNA sensor to detect DNA based on target catalyzed hairpin assembly strategy [148]. Copyright 2014. Reproduced with permission from Elsevier B.V.

**Figure 9 sensors-19-04916-f009:**
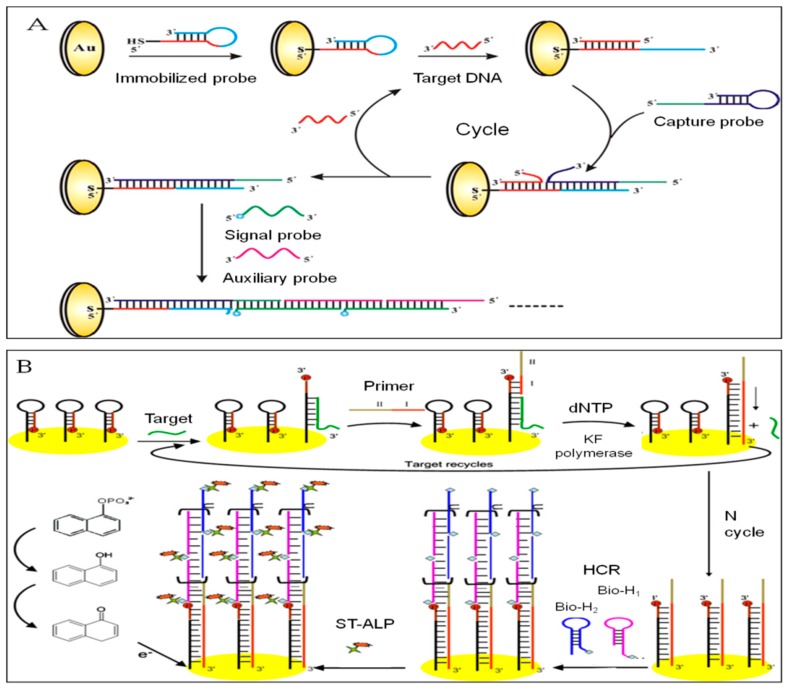
(**A**) An isothermal, enzyme-free and ultrasensitive design to detect DNA based on HCR and DNA catalyzed hairpin assembly (CHA) [154]. Copyright 2013. Reproduced with permission from Elsevier B.V. (**B**) An electrochemical DNA biosensor based on HCR and circular strand-displacement polymerase reaction (CSPR) [155]. Copyright 2013. Reproduced with permission from Elsevier B.V.

**Figure 10 sensors-19-04916-f010:**
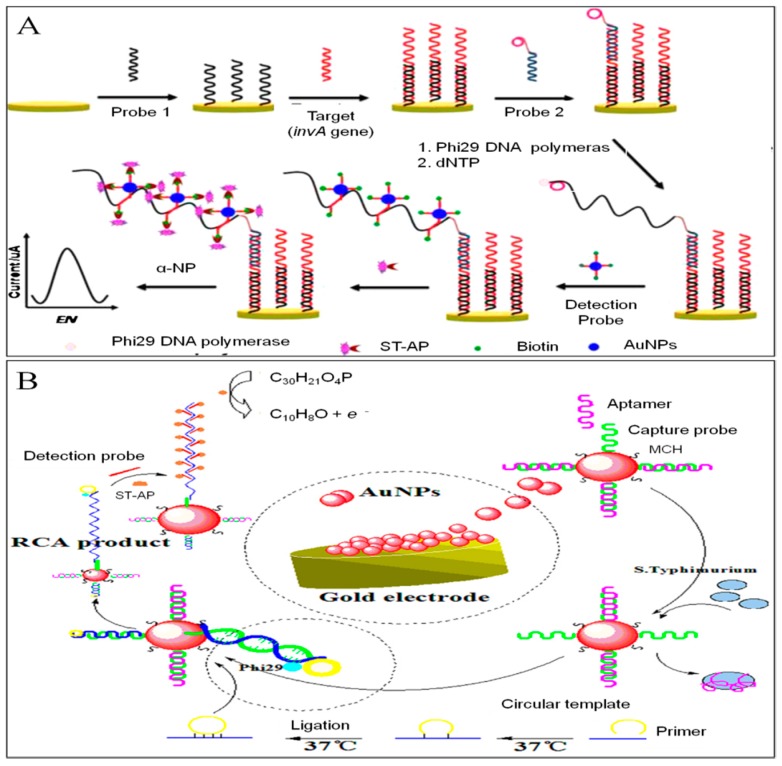
(**A**) A novel electrochemical sensing strategy based on RCA to detect *Salmonella* [81]. Copyright 2014. Reproduced with permission from Elsevier B.V. (**B**) An electrochemical DNA biosensor based on RCA and target-induced aptamer displacement for *S. Typhimurium* detection [100]. Copyright 2018. Reproduced with permission from Elsevier B.V.

**Figure 11 sensors-19-04916-f011:**
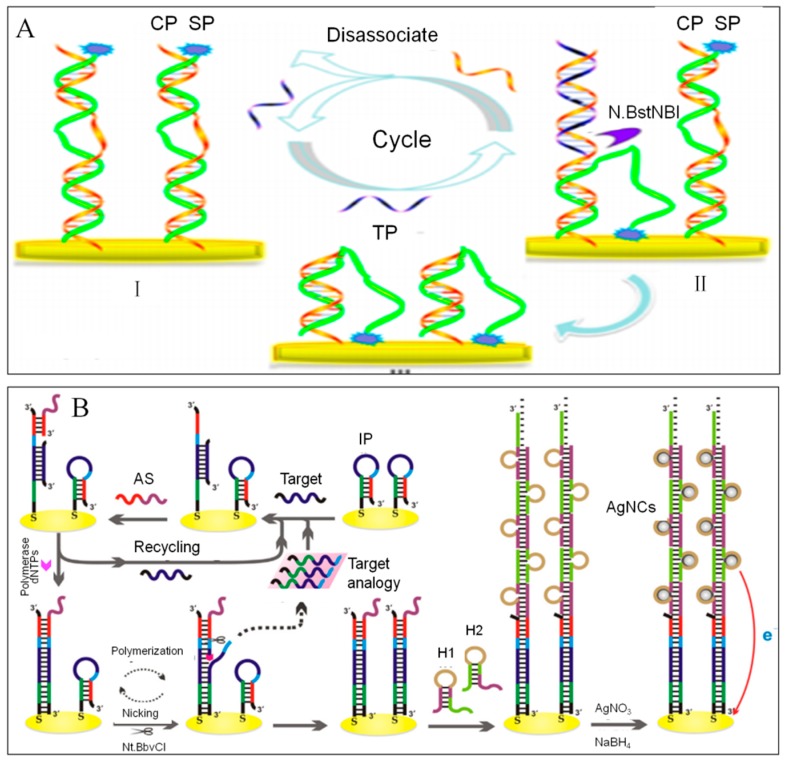
(**A**) An electrochemical DNA biosensor to investigate 16S rDNA of *Bacillus subtilis* based on SDA [119,144]. Copyright 2014. Reproduced with permission from American Chemical Society. (**B**) A label-free and ultrasensitive electrochemical DNA biosensor based on the cascade ASDA and HCR strategy [161]. Copyright 2018. Reproduced with permission from Elsevier B.V.

**Figure 12 sensors-19-04916-f012:**
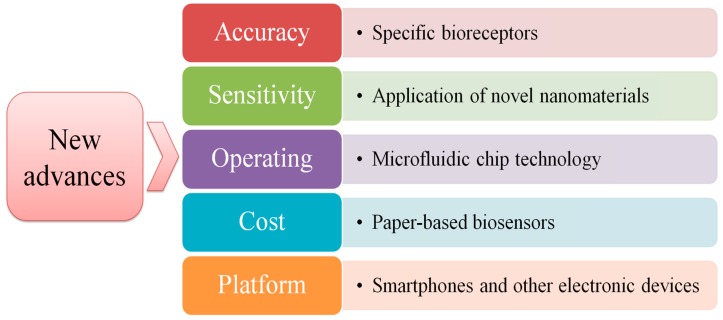
New advances of biosensors.

**Table 1 sensors-19-04916-t001:** Common food-borne pathogens.

Pathogens	Gram	Virulence Factors	Epidemics	Food Source	Refs.
*Salmonella*	-	Enterotoxin	Typhoid fever, paratyphoid fever, gastroenteritis, and septicemia	Egg, raw milk and their products, commercial cold food dishes, raw poultry and meat	[3,4,5,6,7]
*E. coli* O157:H7	-	Endotoxin, exotoxin, capsule, and adhesin.	Acute gastroenteritis and acute dysentery	Meat, fruits, vegetables, commercial cold food dishes, ready-to-eat food, drinking water	[7,8,9,10,11,12,13,14,15]
*Listeria monocytogenes*	+	Endogenous hormone, phagosome, and surface protein	Listeriosis	Frozen food, cheese, milk, meat products, ice, vegetable salad, ready-to-eat food, commercial cold food dishes	[7,16,17,18,19,20,21,22,23]
*Staphylococcus aureus*	+	Hemolytic toxin, leukocidin, enterotoxin, plasma coagulase, and deoxyribonuclease	Suppurative infection, pneumonia, pseudomembranous colitis, pericarditis, sepsis, septicemia	Milk, meat, eggs, fish and their products, commercial cold food dishes	[7,24,25,26]
*Shigella*	-	Endotoxin and exotoxin	Bacterial dysentery	Cooked food and raw material	[27,28,29]
*Cronobacter*	-	Enterotoxin, and adhesion factor	Necrotizing colitis, neonatal meningitis, and bacteremia	Powdered infant formula and milk powder	[30,31,32,33]
*Vibrio parahemolyticus*	-	Hemolysin and urease	Food poisoning, and acute diarrhea	Seafood such as fish, shrimp, crab, shellfish, and seaweed	[34,35,36]
*Proteus*	-	Endotoxin, and heat-resistant enterotoxin	Food poisoning, and acute diarrhea	Food of animal origin, bean products	[37,38]
*Clostridium botulinum*	+	Botulinum toxoid	Muscle relaxation paralysis, and respiratory paralysis	Canned products, cured meat	[39,40,41]
*Bacillus cereus*	+	Enterotoxin	Food poisoning	Leftovers of different meals, commercial cold food dishes	[7,42,43,44]
*Campylobacter*	-	Endotoxin, exotoxin, invasive protein, adhesion, and flagellum	Bacterial gastroenteritis	Raw chicken and by-products	[45,46]

**Table 2 sensors-19-04916-t002:** Current methods for food-borne pathogens detection.

Method	Derivative	Analysis Time	Advantages	Disadvantages	Refs.
Traditional microbiological culture	Chromogenic medium method	5–7 days	High accuracy	Time-consuming, laborious, poor sensitivity and specificity	[56,57]
Immunological method	ELISA, immunomagnetic separation (IMS), immune colloidal gold technique (GICT)	4 h	Rapid, relatively high sensitivity and specificity	High false positive rate and poor stability	[58,59,60,61,62,63]
PCR	Real time-PCR, multiple PCR	≤2 h	Relatively sensitive and rapid, multiple detection	The need of expensive thermal cycle instruments and trained users	[64,65,66,67,68,69]
Nucleic acid-based isothermal amplification assays	LAMP, rolling circle amplification (RCA), saltatory rolling circle amplification (SRCA)	≤2 h	No need for thermal cycle instruments, high sensitivity and selectivity	Not suitable for on-site detection	[70,71,72,73]
Biosensors	Based on signal amplification techniques such as nanotechnology	≤2 h	Rapid, cost-effective, high sensitivity and selectivity	Most cannot achieve multiple detection	[74,75,76,77,78]

**Table 3 sensors-19-04916-t003:** Common methods of DNA immobilization.

Methods	Principle	Evaluation
Adsorption	The skeleton of ssDNA is negatively charged, by modifying the surface of electrodes with positively charged substances or applying a positive potential, DNA can be absorbed on the electrodes.	Simple, with no need of any chemical reagents and DNA probes modification [101]. Low DNA hybridization efficiency.
Covalent binding	DNA is immobilized on the surface of electrodes through the formation of covalent bonds such as amide bonds, ester bonds, ether bonds, Au-S, and Ag-S et al.	Flexible structure, high efficiency of DNA immobilization and hybridization, but with the need of chemical reagents, and with the possibility of non-specific adsorption.
Affinity binding	Avidin is first adsorbed on the surface of the electrode by covalent binding or electrostatic adsorption, and then the biotin-modified DNA is immobilized on the electrode by affinity interaction between biotin and avidin.	The method is simple, stable and resistant to the extreme of temperature, pH, denatured detergents, and organic solvents [101].

**Table 4 sensors-19-04916-t004:** Electrochemical techniques.

Electrodes	Targets	Detection Techniques	Linear Range	LOD	Ref.
Glassy carbon electrode (GCE)	*Salmonella* DNA	CV, EIS, DPV	10–400 and 1–400 pM	2.1 and 0.15 pM	[103]
Gold disk electrode	*Salmonella typhimurium*	CV, DPV	10^2^–10^8^ CFU mL^−1^	3 CFU mL^−1^	[104]
GCE	*Staphylococcus aureus*	CV, EIS	10–10^6^ CFU mL^−1^	10 CFU mL^−1^	[107]
Gold electrode (GE)	*Staphylococcus aureus*	EIS	-	10 CFU mL^−1^	[108]
GE	*Escherichia coli*, *K. pneumoniae*	EIS	10^2^–10^6^ CFU mL^−1^	100 CFU mL^−1^	[109]
GE	*E. faecalis, B. subtilis*	EIS	10^3^–10^6^ CFU mL^−1^	1000 CFU mL^−1^	[109]
GE	*S. aureus*, *E. faecalis*, *P. aeruginosa*, *E. coli* and *Salmonella typhimurium*	CV, EIS	10^1^–10^4^ CFU mL^−1^	10 CFU mL^−1^	[110]
Indium tin oxide (ITO)	*Salmonella typhimurium* DNA	CV, DPV	10 fM–50 nM	10 fM	[111]
ITO	*Escherichia coli* O157:H7 DNA	CV, EIS	1 uM–10 fM	10 fM	[112]
GE	*Bacillus cereus* spore simulant	EIS	10^4^–5 × 10^6^ CFU mL^−1^	3000 CFU mL^−1^	[113]
Carbon paste electrode (CPE)	*Aeromonas hydrophila* DNA	SWV	-	160 fM	[114]
Carbon ionic liquid electrode (CILE)	*Listeria monocytogenes* DNA	CV, EIS, DPV	1 uM–1 pM	290 fM	[85]
Pt/Ir electrodes	*Listeria monocytogenes*	CV, DPV	-	100 CFU mL^−1^	[115]
ITO	*Salmonella typhimurium* DNA	DPV, EIS	4 aM–24 fM	4 aM	[116]
GE	*Enterobacteriaceae* bacteria DNA	SWV, DPV, EIS	0.01 pM–1 nM	8.7 fM	[117]
GE	*Salmonella*	SWV, DPV, EIS	2 × 10^2^–2 × 10^6^ CFU mL^−1^	200 CFU mL^−1^	[118]
GE	*Bacillus subtilis* DNA	DPV	0.1 fM-20 fM	0.08 fM	[119]
GCE	*Salmonella*	CV, EIS	75-7.5 × 10^5^ CFU mL^−1^	25 CFU mL^−1^	[120]
ITO	*Salmonella typhimurium*	CV, EIS	-	10 CFU mL^−1^	[121]
Pencil graphite electrode (PGE)	*Bacillus cereus*	DPV, EIS	10^0^–10^7^ CFU mL^−1^	9.4 pM	[76]
CILE	*Yersinia enterocolitica* DNA	DPV	1 uM–10 PM	1.76 pM	[122]
GCE	*E. coli* O157:H7 DNA	CV, EIS, DPV	-	19.7 fM	[79]
GE	*Salmonella typhimurium*	DPV	72–7.2 × 10^6^ CFU mL^−1^	28 CFU mL^−1^	[123]
GCE	DNA	ASV, EIS	-	100 aM	[124]

- Not available

**Table 5 sensors-19-04916-t005:** Redox active molecules applied in electrochemical DNA biosensors

Redox Active Molecule	Classification	Target	Principle	Refs.
Methylene Blue (MB)	Organic dye	*Bacillus cereus*; *Listeria monocytogenes*	MB covalently interacts with G bases of DNA	[76,85]
Toluidine Blue (TB)	Organic dye	*Enterococcus faecalis*	TB binds to a negatively charged phosphate group	[80]
Oracet Blue (OB)	Organic dye	*Helicobacter pylori*	The hydrophobic rigid plane of OB inserts into the dsDNA base pair	[128,129,131]
Hoechst 33258	Organic dye	*Aeromona hydrophila*	Hoechst 33258 can bind to dsDNA by minor and major groove interaction	[114]
[Ru(phen)_3_]^2+^	Metal complex	*Aeromona hydrophila*	Ru(phen)_3_^2+^ can intercalate into the groove of dsDNA	[97,129]
Daunomycin	Drug molecular	*Aeromona hydrophila*	The molecular carbocyclic moiety can be inserted into the base pair of the DNA helix, and the amino sugar moiety generate electrostatic interaction with the phosphate backbone of the DNA	[114]

**Table 6 sensors-19-04916-t006:** Nanocomposites-based electrochemical DNA biosensors for food borne bacterial pathogen detections.

Nanocomposites/Electrode	Features	Immobilizing Methods of DNA	Targets	LOD (mol/L)	Ref.
AgNCs/AuNPs/GCE	AgNCs are used as direct signal indicator and AuNPs as carrier for signal amplification	By the Au-S bonds between AuNPs and SH-DNA	*Salmonella*	1.62 × 10^−16^	[142]
CTS/V_2_O_5_/MWCN/CILE	Great biocompatibility of V_2_O_5_ nanobelt and excellent electron transfer ability of MWCNTs	CTS can be used for DNA immobilization by electrostatic attraction	*Yersinia enterocolitica*	1.76 × 10^−12^	[122]
NiO/GR/CILE	Graphene and nickel oxide composite possess high surface area and strong affinity with phosphate groups of ssDNA	By the strong affinity between NiO and phosphate groups of ssDNA	*Salmonella enteritidis*	3.12 × 10^−14^	[143]
DpAu/GOx/GCE	GOx has fast electron transfer kinetics and large specific surface area. Thi has good electrochemical redox active properties. Au@SiO_2_ can provide a microenvironment to retain the DNA tag conformation and make them free in orientation	By the Au-S bonds between Au@SiO_2_ and SH-DNA	*E.coli* O157:H7	1.0 × 10^−11^	[144]
Au/GR/CILE	Graphene (GR) possesses high thermal conductivity, good mechanical strength, high mobility of charge carriers, big specific surface area and upstanding electrical properties. The dendritic nanogold provides more sites for the self-assembly of MAA on the electrode surface	By the covalent bonds between the amine groups of ssDNA and the carboxyl group modified on the CILE surface	*Listeria monocyto*	2.9 × 10^−13^	[85]
CTS/Co_3_O_4_/GR/CILE	The nanocomposite film has a very large surface area, good conductivity and excellent porous structure, which lead to the measurable currents even for low concentrations of ssDNA sequence	ssDNA was immobilized on the CTS/Co_3_O_4_/GR/CILE surface by electrostatic attraction	*Staphylococcus aureus*	4.3 × 10^−13^	[145]
AuNPs/CS/MWCNT/AuE	CS–MWCNTs greatly increase effective surface area and electron conductivity. AuNPs provide a biocompatible interface for DNA	By the Au-S bonds between AuNPs and SH-DNA	*Staphylococcus aureus*	3.3 × 10^−16^	[141]
CeO_2_NPs/RGO/GCE	RGO has an extremely large surface area, excellent thermal and electrical conductivity; CeO_2_ possesses high catalytic activity and biocompatibility	By the Π-Π stacking between RGO and DNA bases and electrostatic attraction between CeO_2_NPs and DNA	*Aeromonas hydrophila*	1.0 × 10^−16^	[140]

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
