# Peer review of "Review of Electrochemical DNA Biosensors for Detecting Food Borne Pathogens"

_sensors, 2019, doi:10.3390/s19224916_

Round 1
Reviewer 1 Report
This is a review paper about the most relevant electrochemical DNA biosensors used for the detection of foodborne pathogens. Although it contains interesting information, the paper is poorly structured, badly written and sometimes it is disordered and disorganized. English should be revised. For example, the authors write "The electrodes mainly contain gold electrodes.." Sentences such as "Or its complementary DNA can perform amplify in the presence of targets." or “Because the target can induce dissociation and replacement of the complementary DNA from the aptamer." do not have any sense.
Some figures are slightly blurred.
Some other aspects are the following:
Authors say, “Biosensors are generally constituted of three major components” However, the main components of a typical biosensor are two: the bioreceptor, which specifically interacts with the target analyte, and the transducer, which converts this interaction into an electronic signal. Although they also have an electronic system that includes a signal amplifier, processor, and display.
Section 2.2.1. The type of receptor is only related to aptamers. I miss "Genosensors" or "DNA sensors" in this section.
Section 2.2.1.2 should not be considered as a heading
Section 2.2.2.- Table 3 does not make any sense in this review because it deals with the general methods of immobilization of the receptors and the authors only comment in Section 2.2 the immobilization of thiolated ssDNA on gold modified electrodes by Au-S bonds. For this section to be meaningful, it is necessary to provide other examples with different methods of immobilization in the function of the electrode surface, such as affinity interaction avidin/streptavidin which is another common strategy of probe attachment, electropolymerized monolayers such as 4-aminobenzoic acid, etc.
Section 2.3 title refers to electrochemical characterization and detection methods, doesn't it? Therefore, this Section should be named as Electrochemical techniques because CV, DPV, EIS are technique no methods. For example, “EIS is a frequency domain measurement method with …” is incorrect because EIS is a technique, not a method.
Section 2.4 does not provide any indicators and no example is discussed. In my opinion, indirect electrochemical DNA sensing using enzymes, organic dyes or metal complexes to detect the DNA hybridization event should have been commented in the text.
Moreover, this Section should be renamed as Detection Methods
Free-labeled indicators in Table 5 are redox active molecule. I'd prefer this last title in the first column of Table 5. Some of these examples should be commented on in the text.
Section 2.5 is too short, without details and explanations. Anyway, in my view, this topic should not be a heading although different blocking agents can be mentioned in some examples
One more time, section 3.1.2 is too short. I mean that none of the examples in Table 6 have been commented on in the text.
The sentence "has attracted much attention in the field of electroanalysis because of its advantages of large specific surface area and high electrical conductivity. Therefore, based on these advantages" (lines 269-271) because GO has already introduced in Section 3.3.1.
The meaning of the acronym CNKI (China National Knowledge Infrastructure) is not in the text.
Food borne or foodborne
Unfortunately, and, given these major concerns, I deeply regret not being able to recommend the publication of this paper in Microchemical Journal
Sincerely,
The reviewer
Reviewer 2 Report
This manuscript reviews the use of electrochemical DNA biosensors for the detection of foodborne pathogens. The review is clear and nice. However, the following comments need to be addressed before the manuscript can be accepted for publications.
1.In introduction, the authors should mention the previous relevant reviews and justify the significance of the present review. For example, what are the most closely related reviews in the field and how does the presented paper go beyond them?
2.The model for the nucleic acid-based amplification technology was suggested to draw in order to make the readers easy to understand.
3.The grammar of the manuscript should be polished and the usage license of the figure from other papers should be displayed in legends
Reviewer 3 Report
Dear Authors,
Information provided in this manuscript is overloaded. It can be organized wisely. Please try to organize all the chapters you provided with sequential manner and most important thing is to send a clear massage to reader on electrochemical DNA sensor for food borne pathogens.
In chapter 3, Authors explain about the use of graphene/CNTs and other nanomaterials for amplify the signal of DNA based sensor but there is no explanation on how DNA can be immobilize on surface modified by these materials. In chapter 3.2 (line 279), There is well explanation on nucleic acid-based amplification techniques but lack of connection with electrochemical techniques as title of the manuscript. It seems it is for more fundamental molecular techniques. Please use high resolution image in all figures as all images used in this manuscript has no clarity at all. In chapter 3.2.1.2, Circular strand-replacement polymerization: Authors revealed that targets induced more amplification and amplified largely which induce more electrochemical signal but in figure it seems exactly opposite. Please check it again. It seems when DNAs amplify MB pushes away from the electrode surface then how it give more intense signal?? In chapter 3.2.1.3, Not well explained, how weak binding with duplex help to trigger TMB oxidation?? In chapter 3.2.2, Please explain on how HCR help to amplify MB signal?? which is not clear at all. In future perspectives, Please remove citation from it and try to focus only in electrochemical DNA based sensor
Reviewer 4 Report
The manuscript "A review on electrochemical DNA biosensors for detecting food borne pathogens" describes various methods proposed for the detection of known food pathogens. It contribute to the field of electrochemical biosensors as it resumes various adopted strategies.
I have some comments/ suggestions for the authors:
When reporting the genus ane species the genus is reported in capital letters, the specie in lower case in Italic. Table 1 (check also table 4 and 5): Vibrio parahaemolyticus instead of Vibrio Parahaemolyticus.
As reported by Vizzini et al. Micromachines 2019, 10, 500; doi:10.3390/mi10080500, Campylobacter is one of the most important food pathogens (according to EFSA data). It should be included in the list.
Lane 29: Listeria monocytogenes instead of Listeria Monocytogenes
check references as genus and species are often written not in Italic; whereas Salmonella Typhimurium instead of Salmonella Typhimurium as Typhimurium is a serovar not a specie.
Lane 45: add the meaning ofPCR, LAMP and RCA
Lane 110: the text report the analyses of meat, but the value is expressed in CFU/mL, check if the anlyses was conducted ona solid or liquid sample
Lane 148: which is the sensitivity of the method described by Zahra Izardi et al?
Lane 153: "Modifying the amino group at the 5' end of aptamer before immobilizing on electrode surface." please describe which modification was proposed.
Check the position of the references in the text. Table 5 is described in paragraph 2.4 and shows references from 134,135, 136
Paragraph 2.2.1 can be improved, a more detailed description of the DNA based bioreceptor can be added.
Moreover a description of the label-free methods and methods which use labeled probes should be reported, and also related advantages /disadvantages.
in "Summary and conclusions" the authors cite portability of the biosensors. Few comments on this aspect could be added related to the described biosensors.
Round 2
Reviewer 1 Report
Dear Editor,
This review summarized recent electrochemical DNA sensors for the determination of foodborne pathogens. Although the revision made by the authors, addressing all the reviewer’s comments, has improved the quality of this paper, I still think that there are some points that need to be addressed before its publication in Sensors.
1) Although the paper contains a great amount of information, as corresponds to a Review article, in my opinion, the following references should be cited:
- B. Rafique, M. Iqbal, T. Mehmood and M.A. Shaheen. Electrochemical DNA biosensors: a review. Sensor Review, https://doi.org/10.1108/SR-08-2017-0156
- J.I.A. Rashid, N.A. Yusof. The strategies of DNA immobilization and hybridization detection mechanism in the construction of electrochemical DNA sensor: A review. Sensing and Bio-Sensing Research 16 (2017) 19–31.
2) As I said in my previous revision, the authors claim that a biosensor has three main components (line 56) and I state that the main components of a typical biosensor are two: the bioreceptor, which specifically interacts with the target analyte, and the transducer, which converts this interaction into an electronic signal. Although they also have an electronic system that includes a signal amplifier, processor, and display.
3) In my opinion, it's not appropriate to talk about techniques on the 80 line. I would prefer to change the words "common techniques" to "strategies" and name Section 3 as "Strategies for improving the sensitivity of electrochemical DNA biosensors"
On the other hand, in my mind, it is not uncommon to comment on strategies based on the use of functional nanomaterials to improve sensitivity (lines 80-82). What is less common is to summarize the most used nucleic acid amplification technologies in order to improve the limit of detection. Therefore, this sentence should be rewritten.
4) In my view, self-assembling of thiolated-DNA or aptamer on gold surfaces is an immobilization protocol that could be included in the covalent binding methods (2nd paragraph section 2.2.2 and Table 3.
Effectively, most DNA immobilization methods on gold electrodes are based on the use of thiolate DNA probes. That is why authors have to specify that in the Principle column, as they do in in the Evaluation one.
5) An example of indirect DNA hybridization detection using Ag nanoparticle labels by ASV technique could be included in Table 4
6) Table 3.- Potentiostatic adsorption and Electrostatic adsorption: In the first case, you have to apply a potential and in the second one you do the same but at open circuit potential, right? In my view, both could be merged and named as “Adsorption”.
7) Section 2.4.- For better understanding, DNA hybridization detection methods should be divided into label-free and label-based methods and the label-based methods in those based on the incorporation of redox active indicator, enzyme label or nanoparticles to the DNA.
8) Section 3.2.- Some subsections of this section are written in a confusing way. A clearer explanation is suggested to make them easier for readers to understand. Perhaps, some figures could also be adapted.
In contrast, other subsections such as Section 3.2.1.3 now is well understood but an example of this strategy should appear in the manuscript
Sincerely,
The reviewer
Reviewer 2 Report
The revised manuscript has been modified according to my suggestions. I think this version has been improved and could be considered accpted by this journal.
Author Response
I’m truly grateful for your help!
Reviewer 3 Report
Still Figure 9,10 and 11 have very low resolution which make these figures very less understandable to readers. Please replace these figures with high resolution. You can easily draw these figures with high resolution format.
Author Response
Thanks for your suggestions. We have replaced these figures with high resolution in the revised version.
Round 3
Reviewer 1 Report
Dear Editor,
The authors present a review of the most relevant electrochemical DNA biosensors used for the detection of foodborne pathogens. The authors' review, taking into account the reviewers' comments, has improved the quality of this paper. I now feel confident to support its publication in Sensors, but it should correct some minor error:
- The word “capture” is misspelt on line 278 and genosensor on line 280
Sincerely,
The reviewer
Author Response
Point 1: The word “capture” is misspelt on line 278 and genosensor on line 280
Response 1: We are sorry for our incorrect writing. These two words have been corrected in the revised version (line 278 and 280, page 10).
Special thanks to you for your good comments.